# SIRT6 Inhibits Anoikis of Colorectal Cancer Cells by Down-Regulating NDRG1

**DOI:** 10.3390/ijms25115585

**Published:** 2024-05-21

**Authors:** Fengying Li, Wentao Yu, Xiaoling Zhou, Jingyu Hou, Yunyi Gao, Jun Zhang, Xiangwei Gao

**Affiliations:** 1Sir Run Run Shaw Hospital, School of Public Health, Zhejiang University School of Medicine, Hangzhou 310058, China; lify@zju.edu.cn; 2Department of Clinical Laboratory, Sir Run Run Shaw Hospital, Zhejiang University School of Medicine, Hangzhou 310016, China; 3Environmental Medicine, Zhejiang University School of Medicine, Hangzhou 310058, China; 22218244@zju.edu.cn (W.Y.); 22118834@zju.edu.cn (X.Z.); jingyuhou@zju.edu.cn (J.H.); yunyigao@zju.edu.cn (Y.G.)

**Keywords:** colorectal cancer, anoikis, SIRT6, epigenetic regulation, NDRG1

## Abstract

Anoikis, a form of apoptosis resulting from the loss of cell–extracellular matrix interaction, is a significant barrier to cancer cell metastasis. However, the epigenetic regulation of this process remains to be explored. Here, we demonstrate that the histone deacetylase sirtuin 6 (SIRT6) plays a pivotal role in conferring anoikis resistance to colorectal cancer (CRC) cells. The protein level of SIRT6 is negatively correlated with anoikis in CRC cells. The overexpression of SIRT6 decreases while the knockdown of SIRT6 increases detachment-induced anoikis. Mechanistically, SIRT6 inhibits the transcription of N-myc downstream-regulated gene 1 (*NDRG1*), a negative regulator of the AKT signaling pathway. We observed the up-regulation of SIRT6 in advanced-stage CRC samples. Together, our findings unveil a novel epigenetic program regulating the anoikis of CRC cells.

## 1. Introduction

Colorectal cancer (CRC) is one of the most prevalent malignancies worldwide, ranking third in terms of global incidence and second in terms of mortality among all malignant tumors [1]. Despite advancements in early CRC diagnosis and treatment, approximately 21% of cases are diagnosed at an advanced stage characterized by metastasis, with a mere 14% five-year survival rate [1]. Given that metastasis is the leading cause of mortality in CRC patients, comprehending the mechanisms driving CRC metastasis is essential for enhancing patient survival and prognosis.

The growth of normal epithelial cells relies on their attachment to the extracellular matrix (ECM). Cell–ECM adhesion activates diverse signaling pathways, without which cells undergo a form of programmed cell death known as “anoikis” [2]. Anoikis acts as a natural defense mechanism against cancer cell metastasis. However, malignant tumor cells acquire resistance to anoikis, enabling their survival in the bloodstream and the subsequent formation of metastatic lesions in distant organs [3]. Cancer cells achieve anoikis resistance by activating pro-survival signaling pathways such as the phosphatidylinositol 3-kinase (PI3K)/protein kinase B (AKT) pathway [4,5,6]. It was suggested that upon ECM detachment, cancer cells adopt rounded morphologies and form small hemispherical plasma membrane protrusions called blebs, while sustained membrane blebbing promotes the activation of pro-survival signaling [7].

The induction of anti-apoptotic genes is attributed to anoikis resistance in cancer cells. Recently, the role of epigenetic regulation, such as histone modifications, in regulating gene transcription during ECM detachment has gained much attention [8]. For instance, it has been demonstrated that lysine-specific histone demethylase 3A (KDM3A), a histone H3 demethylase, promotes anoikis resistance by activating the expression of the pro-apoptotic genes Bcl-2 19-kDa-interacting protein 3 (*BNIP3*) and BNIP3-like (*BNIP3L*) [9]. Overall, our understanding of the epigenetic alterations in detached cancer cells remains limited, and the key regulatory factors remain to be identified.

SIRT6 is a member of the silent mating type information regulation 2 homolog (sirtuin) family, which exhibits deacetylase activity and single ADP ribosyl transferase activity [10,11]. Mainly located in the nucleus, SIRT6 plays a critical role in regulating genome stability, gene expression, and DNA repair by deacetylating histones [12]. The dysregulation of SIRT6 is associated with metabolic abnormalities and degenerative pathologies [13,14]. Reduced SIRT6 expression has been observed in various human cancers, and its expression level has been positively correlated with the prognosis of cancer patients [15,16,17]. Studies in mice have shown that the loss of SIRT6 promotes the development of colorectal cancer and pancreatic cancer [15,18]. Nevertheless, SIRT6 has also been reported to promote cancer progression in skin cancer, prostate cancer, and breast cancer [19,20,21,22]. These studies suggest that SIRT6 might play complex roles in tumorigenesis and progression, depending on stages and cancer types. Specifically, the potential involvement of SIRT6 in cancer cell anoikis remains unexplored.

In this study, we observed a negative correlation between the expression of SIRT6 and anoikis in CRC cells. Subsequent experiments elucidated a role of SIRT6 in promoting anoikis resistance. We identified the *NDRG1* gene, a negative regulator of the AKT signaling pathway, as a direct target of SIRT6. Our findings indicate that targeting SIRT6 could be a potential therapeutic strategy for the treatment of CRC.

## 2. Results

### 2.1. The Protein Level of SIRT6 Is Decreased during the Anoikis of CRC Cells

To investigate the role of SIRT6 in the anoikis process of colorectal cancer cells, we cultured RKO cells in suspension and assessed the expression of SIRT6. Immunoblotting and immunofluorescence experiments revealed a significant reduction in SIRT6 protein levels after 12 h of ECM detachment (Figure 1A,B). However, the mRNA level of the *SIRT6* gene remained largely unchanged (Figure 1C), implying that regulation happens at the post-transcriptional level. We therefore examined the impact of ECM detachment on SIRT6 protein stability. To that end, we inhibited protein synthesis using cycloheximide and determined the remaining SIRT6 protein level. The degradation of the SIRT6 protein accelerated in detached cells in comparison to attached cells (Figure 1D), suggesting that ECM detachment predominantly promotes the degradation of the SIRT6 protein.

### 2.2. The Protein Level of SIRT6 Is Higher in Anoikis-Resistant CRC Cells

We compared SIRT6 expression in two isogenic colon carcinoma cell lines with varying anoikis potential: SW480 cells and SW620 cells. SW480 cells were derived from primary colon carcinoma, whereas SW620 cells were derived from metastatic cancer cells in the mesenteric lymph node of the same patient. As anticipated, Annexin V staining data showed that SW480 cells displayed anoikis sensitivity, while SW620 cells exhibited anoikis resistance (Figure 2A). Immunoblotting experiments revealed a significantly higher SIRT6 protein level in SW620 cells in comparison to SW480 cells (Figure 2B). Moreover, the protein stability was much higher in SW620 cells than in SW480 cells (Figure 2B). These data indicate a positive correlation between SIRT6 protein level and anoikis resistance in CRC cells.

### 2.3. SIRT6 Inhibits the Anoikis of CRC Cells

To investigate the role of SIRT6 in CRC cell anoikis, we plated both control cells and SIRT6-knockdown cells in ultra-low attachment plates for 36 h and subsequently analyzed apoptosis using Annexin V staining. Our data showed that the knockdown of SIRT6 did not induce apoptosis under normal growth conditions. However, a significant increase in apoptosis was observed in SIRT6-deficient cells compared to control cells under ECM detachment culture conditions (Figure 3A–C). To consolidate the role of SIRT6 in anoikis, we established stable cell lines overexpressing SIRT6. Conversely, the overexpression of SIRT6 significantly reduced anoikis in RKO cells (Figure 3D–F). These results collectively support the conclusion that SIRT6 promotes CRC cell anoikis resistance.

### 2.4. Identification of SIRT6-Regulated Gene Expression

To investigate the mechanisms underlying the anti-anoikis effects of SIRT6, we conducted a transcriptomic analysis in cells overexpressing SIRT6 and cells with SIRT6 knockdown. Employing a fold change threshold of 2-fold, we identified 405 down-regulated genes and 319 up-regulated genes in SIRT6 overexpression cells (Figure 4A, Appendix A). Among them, the carbohydrate sulfotransferase 13 (*CHST13*) gene exhibits the most pronounced up-regulation, while the nuclear pore complex-interacting protein family member B9 (*NPIPB9*) gene exhibits the most pronounced down-regulation. Conversely, in SIRT6 knockdown cells, we observed 482 up-regulated genes and 231 down-regulated genes (Figure 4A, Appendix A). Among them, the matrix metallopeptidase 10 (*MMP10*) gene exhibits the most pronounced up-regulation, while the Rho-related BTB domain-containing 3 (*RHOBTB3*) gene exhibits the most pronounced down-regulation. Notably, the RNA-seq data showed a greater number of down-regulated genes than up-regulated genes in SIRT6 overexpression cells and displayed the opposite trend in SIRT6 knockdown cells. These findings align with previous reports indicating that SIRT6 primarily suppresses gene transcription by deacetylating histones [23,24].

We performed gene ontology (GO) analysis and Kyoto Encyclopedia of Genes and Genomes (KEGG) analysis for the differentially expressed genes influenced by SIRT6. We first analyzed the down-regulated genes after SIRT6 overexpression (405 genes). Cell adhesion was the most representative type of biological process that was significantly enriched, such as homophilic cell adhesion via plasma membrane adhesion molecules (GO:0007156), cell adhesion (GO:0007155), and cell–matrix adhesion (GO:0007160) (Figure 4B). The most affected cellular components were associated with the cell membrane, such as the brush border membrane (GO:0031526), microvillus membrane (GO:0031528), and plasma membrane (GO:0005886) (Figure 4B). Concerning molecular functions, calcium ion binding (GO:0005509), metal ion binding (GO:0046872), carbohydrate binding (GO:0030246), and glycosaminoglycan binding (GO:0005539) were mostly enriched (Figure 4B). KEGG analysis revealed the enrichment of tryptophan metabolism (hsa00380) (Figure 4D).

Next, we annotated the function of the up-regulated genes upon SIRT6 knockdown (482 genes). They are enriched in biological processes such as signal transduction (GO:0007165), the positive regulation of interferon–gamma production (GO:0032729), and cell adhesion (GO:0007155) (Figure 4C). Changes in cell components were mainly enriched in the plasma membrane (GO:0005886) (Figure 4C). Molecular functions were mainly enriched in phospholipid binding (GO:0005543), calcium channel regulator activity (GO:0005246), and transmembrane transporter activity (GO:0022857) (Figure 4C). KEGG pathway analysis showed that these SIRT6-regulated genes were primarily enriched in cytokine–cytokine receptor interaction (hsa04060), tyrosine metabolism (hsa00350), the JAK-STAT signaling pathway (hsa04630), and the cAMP signaling pathway (hsa04024) (Figure 4D).

### 2.5. SIRT6 Represses the Transcription of the NDRG1 Gene

We selected the *NDRG1* gene for further investigation due to its significant alteration upon SIRT6 overexpression and knockdown (Figure 5A). We investigated the regulatory role of SIRT6 in *NDRG1* gene transcription. The knockdown of SIRT6 significantly increased NDRG1 expression at both the mRNA and protein levels (Figure 5B,D). Conversely, the overexpression of SIRT6 significantly decreased NDRG1 expression (Figure 5C,D). NDRG1 is reported to be a negative regulator of the AKT signaling pathway [25]. Indeed, we observed increased AKT phosphorylation in SIRT6 overexpression cells and decreased AKT phosphorylation in SIRT6 knockdown cells (Figure 5D). These data indicate that SIRT6 negatively regulates NDRG1 expression, resulting in AKT activation.

To ascertain whether SIRT6 directly binds to the *NDRG1* gene promoter, we conducted chromatin immunoprecipitation (ChIP) assays followed by qPCR analysis. We observed enrichment of the *NDRG1* gene promoter in the anti-SIRT6 group when compared to the IgG control group (Figure 5E). Notably, the *ACTB* gene showed no enrichment in the anti-SIRT6 group, underscoring the specificity of the binding between SIRT6 and the *NDRG1* gene promoter (Figure 5E). SIRT6 is known to deacetylate histone H3 at lysine 9 (H3K9) [23,24]. To investigate whether SIRT6 deacetylates histone H3 at the *NDRG1* gene promoter, we assessed the acetylation of H3K9 (Ac-H3K9) in the *NDRG1* gene promoter region using ChIP. Immunoblotting analysis of the immunoprecipitated products by Ac-H3K9 revealed that the total Ac-H3K9 level did not change in the SIRT6 overexpression group (S6-OE) and SIRT6 knockdown group (S6-KD) (Figure 5F). The data showed that the knockdown of SIRT6 significantly increased while the overexpression of SIRT6 decreased the Ac-H3K9 level in the *NDRG1* gene promoter region (Figure 5G). As a negative control, the Ac-H3K9 levels of the *ACTB* gene remained unchanged (Figure 5G). Collectively, our findings indicate that SIRT6 negatively regulates *NDRG1* gene transcription by directly deacetylating histone H3 acetylation in the *NDRG1* gene promoter region.

### 2.6. Down-Regulation of NDRG1 Contributes to SIRT6-Inhibited Anoikis

It has been reported that NDRG1 suppresses the migration, invasion, and epithelial–mesenchymal transition of CRC, while its function in cell anoikis has not been studied [26]. We investigated the role of NDRG1 in CRC cell anoikis. We observed increased AKT phosphorylation in NDRG1 knockdown cells (Figure 6A). Moreover, the knockdown of NDRG1 inhibited CRC cell anoikis (Figure 6B). To study the causal relationship between NDRG1 down-regulation and SIRT6-inhibited anoikis, we performed a rescuing experiment. The re-expression of NDRG1 in SIRT6-overexpression cells decreased AKT phosphorylation and promoted CRC cell anoikis (Figure 6C,D). Collectively, these results indicated that SIRT6 inhibits anoikis, at least partially, through the down-regulation of NDRG1.

### 2.7. SIRT6 Expression Is Elevated in Advanced-Stage CRC Samples

We further assessed the protein expression of SIRT6 in cancer tissues and paired normal tissues in a cohort of 75 cases of CRC patients. The protein levels of SIRT6 showed no significant change between adjacent normal tissues and CRC tissues (Figure 7A,B). However, the protein levels of SIRT6 were much higher in stage III–IV CRC tissues than stage I–II CRC tissues (Figure 7A,C). These findings in patient samples align with our discovery that SIRT6 inhibits the anoikis of CRC cells and indicates a role of SIRT6 in promoting CRC progression.

## 3. Discussion

SIRT6 plays pivotal roles in cancer-associated pathways, such as maintaining genomic stability, inhibiting cell proliferation, and regulating energy metabolism. In this study, we have demonstrated the involvement of SIRT6 in regulating the anoikis of CRC cells. We observed a decrease in SIRT6 protein levels during anoikis and a higher level of SIRT6 protein in anoikis-resistant CRC cells. The depletion of SIRT6 promotes anoikis, whereas the overexpression of SIRT6 inhibits anoikis. Collectively, our findings suggest that SIRT6 functions as an onco-protein during CRC progression by inhibiting cell anoikis.

The role of SIRT6 in CRC seems to be multifaceted, as certain studies support its cancer-suppressive function, while others indicate a tumor-promoting effect. The down-regulation of SIRT6 has been observed in CRC, and reduced SIRT6 expression is closely linked to carcinogenesis and poor prognosis in CRC [15]. Depletion of the *Sirt6* gene promotes the in situ development of colorectal cancer [15]. The knockdown of SIRT6 abolishes apoptotic responses and confers resistance to chemo-treatment [27]. These reports suggested that SIRT6 may function as a tumor suppressor in CRC. In contrast, Geng et al. reported that the overexpression of SIRT6 accelerates the invasion of colorectal cancer cells [28]. Our data align with this report, suggesting that SIRT6 may promote CRC progression. The intricate role of SIRT6 in CRC echoes the pleiotropic nature of TGF-β, which acts as a tumor suppressor in early-stage tumors but an oncogene in advanced cancer [29]. The complex role of SIRT6 may be modulated by the cellular context and its integration with other signaling pathways.

The reduced protein levels of SIRT6 observed in advanced-stage CRC tissues and highly metastatic SW620 cells highlight its potential as an ideal marker for predicting the metastatic potential of CRC cells. Interestingly, our data suggest that, contrary to changes in mRNA levels, the protein level of SIRT6 was altered in the anoikis process. Notably, SIRT6 protein degradation has been previously reported. Specifically, SIRT6 undergoes phosphorylation at serine338 (Ser338) by the kinase AKT1, leading to its interaction with and ubiquitination by mouse double minute 2 (MDM2), ultimately targeting SIRT6 for protease-dependent degradation [30]. Ubiquitin-specific peptidase 10 (USP10), an α-ubiquitin-specific peptidase, counteracts SIRT6 ubiquitination, thus protecting SIRT6 from proteasomal degradation [31]. These findings collectively support the notion that SIRT6 protein is regulated through ubiquitin–proteasome degradation. Hence, the evaluation of SIRT6 expression should consider the protein level rather than the mRNA level. Further investigations are warranted to elucidate the mechanisms underlying SIRT6 degradation in the context of anoikis.

In this report, we have identified SIRT6 as a negative regulator of *NDRG1* gene transcription. SIRT6 de-acetylates the H3K9 level in the *NDRG1* gene promoter region, resulting in *NDRG1* gene silencing under normal conditions. During the process of matrix detachment, the SIRT6 protein level is decreased, which loosens *NDRG1* chromatin and activates *NDRG1* gene transcription. Accumulating evidence has demonstrated NDRG1 to be a metastasis suppressor in colorectal cancer. NDRG1 is reported to be a favorable predictor for the prognosis of CRC [32]. NDRG1 inhibits the migration, invasion, and epithelial–mesenchymal transition (EMT) of CRC cells [26]. As a metastasis suppressor, NDRG1 regulates several signaling pathways, such as AKT signaling and epidermal growth factor receptor (EGFR) signaling. In our study, the tumor-suppressive function of NDRG1 in CRC was confirmed for silencing NDRG1 expression, leading to enhanced anoikis. More importantly, we found that NDRG1 down-regulation mediated the function of SIRT6 because NDRG1 overexpression almost totally abolished SIRT6-inhibited AKT activation and anoikis.

In summary, our data indicate an essential role of SIRT6 in promoting the anoikis resistance of CRC cells. SIRT6 tends to suppress CRC tumorigenesis at early stages but promote CRC progression at advanced stages. Understanding this conversion is critical for the development of therapeutics that target SIRT6.

## 4. Materials and Methods

### 4.1. Cell Culture and Treatment

RKO, SW480, and SW620 cells were obtained from the American Type Culture Collection (ATCC) and routinely cultured in Dulbecco’s Modified Eagle Medium (DMEM) containing 10% fetal bovine serum at 37 °C and 5% CO_2_. To induce anoikis, cells were cultured in ultra-low attachment plates (Corning, Shanghai, China) for the indicated time and subjected to the following experiments.

### 4.2. Quantitative Polymerase Chain Reaction (PCR) Method (qPCR)

Cells were lysed with Trizol (Invitrogen, Shanghai, China) (1 ml/10 cm^2^), and RNA was extracted according to the Trizol instructions. One microgram of total RNA was subjected to a reverse transcription reaction using ABI’s reverse transcription kit. The SYBR Green method was used for qPCR analysis to detect the expression level of each gene. The expression level of each gene was normalized to the housekeeping gene *β-actin* (*ACTB*). Relative gene expression was calculated and analyzed using the 2^−ΔΔCt^ method.

The primers sequences were as follows: *SIRT6* gene (forward: AGTCTTCCAGTGTGGTGTTCC; reverse: CGTGGGGACCCCTGAAGTC); *NDRG1* gene (forward: AGCTCGTCAGTTCACCATCC; reverse: CGATGTCCTGCTCCTAACGC); *ACTB* gene (forward: ACAGAGCCTCGCCTTTGCC; reverse: GATATCATCATCCATGGTGAGCTGG).

### 4.3. Immunoblotting

Protein samples were denatured, separated by sodium dodecyl sulfate–polyacrylamide gel electrophoresis (SDS-PAGE), and transferred to a polyvinylidene difluoride (PVDF) membrane (Millipore, Burlington, MA, USA). The membrane was blocked, incubated with SIRT6 antibody (Abcam, 1:1000) or NDRG1 antibody (1:1000, Proteintech, Wuhan, China) overnight, and then incubated with horseradish peroxidase (HRP)-labeled secondary antibody at room temperature for 1 h. The membrane was detected by chemiluminescence (GE Technology, Cincinnati, OH, USA).

### 4.4. SIRT6 Protein Degradation Assay

To determine the degradation of the SIRT6 protein, the attached or detached cells were treated with cycloheximide (CHX, 100 μg/mL, MedChemExpress, Shanghai, China) to inhibit protein synthesis for 0 h, 4 h, and 8 h. The remaining SIRT6 protein level was determined by immunoblotting.

### 4.5. SIRT6 Knockdown or Overexpression

To knock down the expression of SIRT6, short hairpin RNAs (shRNA) targeted to the human *SIRT6* gene (shRNA#1: TGGAAGAATGTGCCAAGTGTA; shRNA#2: CAAGTGTAAGACGCAGTACGT) were cloned into the pLKO.1 vector. To knock down the expression of NDRG1, short hairpin RNAs (shRNA) targeted to the human *NDRG1* gene (shNDRG1: CCTGGAGTCCTTCAACAGTTT) were cloned into the pLKO.1 vector. To overexpress SIRT6 levels, full-length human *SIRT6* cDNA was cloned into the pCDH vector. To overexpress NDRG1 levels, full-length human *NDRG1* cDNA was cloned into the pCDH vector. The lentiviral plasmid expressing the target gene or shRNA, pVSV-G, and pCMVR8.9 were co-transfected into HEK293 cells and the supernatant was collected 48 h after transfection. CRC cells were infected with virus-containing supernatant. Seventy-two hours after virus infection, 1 μg/mL puromycin was added for selection. The positive cells were obtained and validated by quantitative PCR and immunoblotting analysis.

### 4.6. Apoptosis Analysis

Cells were cultured in normal dishes or ultra-low attachment dishes for 24 h and then collected. A 100 μL binding buffer was added to resuspend the cell pellet. Amounts of 2.5 μL fluorescein isothiocyanate (FITC)-labeled Annexin V (Annexin V-FITC) and 5 μL propidium iodide (PI) were added to label the apoptotic cells. Flow cytometry was applied to detect apoptosis.

### 4.7. RNA Sequencing (RNA-Seq) Analysis

Total RNA from control, SIRT6 knockdown, and SIRT6 overexpression cells were isolated utilizing the TRIzol reagent. RNA samples were then submitted to mRNA library construction (KaiTai Biotech, Nanning, China) and sequencing by using an Illumina NovaSeq 6000 (San Diego, CA, USA). Illumina sequencing reads were initially processed using fastp (version 0.23.2) for adaptor removal and quality trimming [33]. The processed reads were aligned to the human reference genome (gencode.v41) using Bwa-mem2 (version 2.2.1) with default parameters. SAMtools (version 1.9) was applied to eliminate PCR duplications. Gene expression levels were normalized using the reads per kilobase per million (RPKM) method to account for differences in sequencing depth and gene length. Differential genes were filtered based on statistical significance using the following criteria: log2 fold change greater than 1, adjusted *p*-values (adj *p*-val) less than 0.05, and false discovery rate (FDR) less than 0.05. These thresholds ensured the selection of genes with significant differential expression while minimizing false positives.

Differential genes that met the above criteria were subjected to functional analysis using the Database for Annotation, Visualization and Integrated Discovery (DAVID) database.

### 4.8. Chromatin Immunoprecipitation (ChIP)

RKO cells were incubated with formaldehyde to crosslink the protein–DNA complexes, and the cross-linking was terminated by 0.125 M glycine. SDS lysis buffer was used to lyse the cells, which were then sonicated into DNA fragments, and the supernatant was collected by centrifugation. One part of the sample was retained as the input of the experiment for direct de-crosslinking. The remaining sample was used to incubate with 2 μg of SIRT6 or control IgG at 4 °C overnight. The immune complex was precipitated with Protein A-Agarose beads, and then washed and eluted with lysis buffer and de-crosslinked with 0.2 M NaCl at 65 °C overnight. After a 30 min water bath with RNase A at 37 °C, and 60 min with proteinase K at 45 °C, DNA fragments were recovered by a Tiangen recovery kit (Tiangen Biotech, Beijing, China) and detected by RT-qPCR.

### 4.9. Immunohistochemistry (IHC)

The colorectal cancer tissue arrays were purchased from Shanghai Outdo Biotech Company (Shanghai, China) (HColA150CS02). All subjects gave their informed consent for inclusion before they participated in this study. This study was conducted in accordance with the Declaration of Helsinki, and the protocol was approved by the Ethics Committee of Shanghai Outdo Biotech Company (project identification code: YBM-05-02; Date, 01/2010).

Following deparaffinization and hydration of the tissue array sections, heat-induced antigen retrieval was performed using a citrate buffer solution. Subsequently, the sections were treated with a 3% H_2_O_2_ solution to quench endogenous peroxidase activity and 5% goat serum to block nonspecific binding. The sections were incubated with anti-SIRT6 (Abcam (Cambridge, UK), 1:100) antibodies overnight at 4 °C, followed by incubation with HRP-conjugated secondary antibodies. Staining was achieved using diaminobenzidine as the enzyme substrate, and hematoxylin serving as the counterstain. The slides were digitally scanned using an Olympus digital slicing scanner VS200 (Olympus Corporation, Tokyo, Japan).

### 4.10. Statistical Analysis

A two-tailed Student’s *t*-test was used for the statistical analysis of qPCR data and apoptosis data. A Chi-Square test was used for the statistical analysis of IHC data. *p* < 0.05 was regarded as statistically significant.

## Figures and Tables

**Figure 1 ijms-25-05585-f001:**
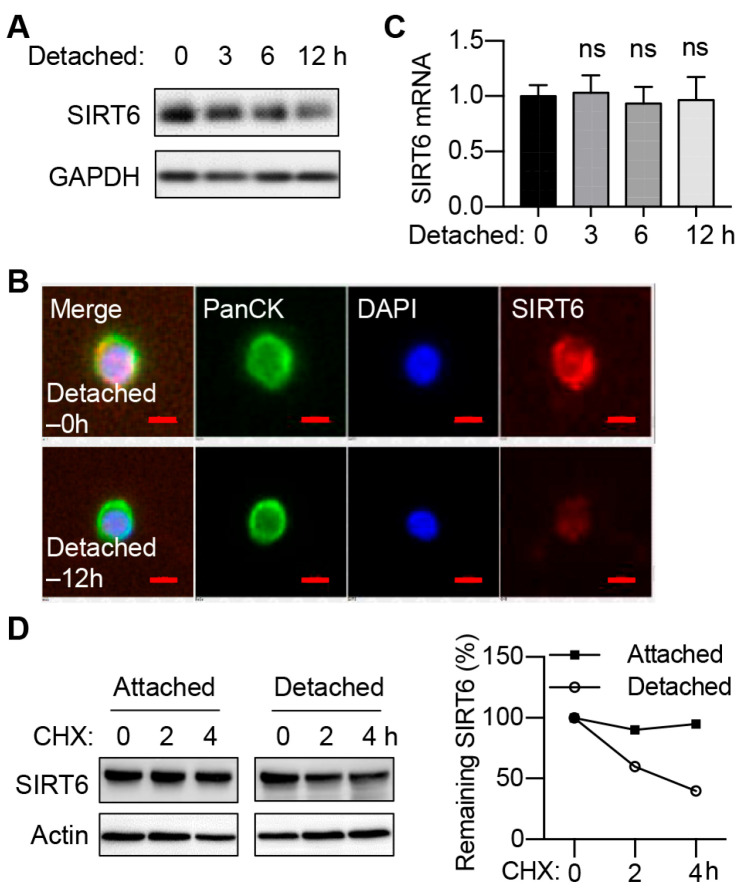
**The protein level of SIRT6 is decreased during the anoikis of CRC cells.** (**A**) The protein level of SIRT6 in RKO cells cultured in ultra-low attachment plates (Detached) for the indicated time. (**B**) Immunofluorescence analysis of SIRT6 protein in RKO cells cultured in ultra-low attachment plates (Detached) for the indicated time. Pan-cytokeratin (PanCK) was used as an epithelial marker. Scale bar, 5 μm. (**C**) The mRNA level of the *SIRT6* gene in RKO cells cultured in ultra-low attachment plates (Detached) for the indicated time. A two-tailed *t*-test was used for statistical analysis. ns—not significant. (**D**) The degradation of the SIRT6 protein in RKO cells cultured in normal dishes (Attached) or ultra-low attachment dishes (Detached) for 12 h. Attached and detached cells were treated with cycloheximide (CHX, 100 μg/mL) to block protein synthesis. The remaining SIRT6 protein level was determined by immunoblotting.

**Figure 2 ijms-25-05585-f002:**
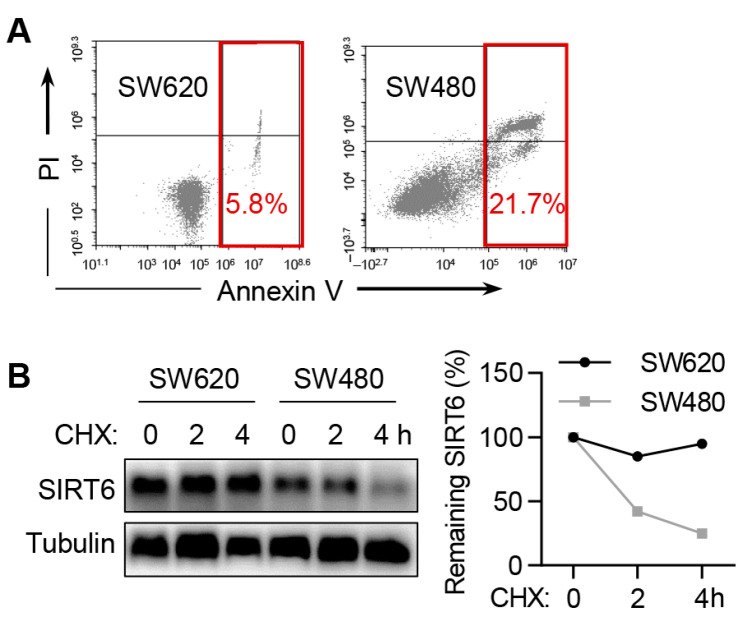
**The protein level of SIRT6 is higher in anoikis-resistant CRC cells.** (**A**) The apoptosis levels of SW480 cells and SW620 cells cultured in ultra-low attachment dishes for 48 h. Cell apoptosis was monitored by annexin V-FITC/PI staining and flow cytometry. (**B**) The degradation of SIRT6 protein in SW480 cells and SW620 cells cultured in ultra-low attachment dishes for 24 h. Cells were treated with CHX (100 μg/mL) to block protein synthesis, and the remaining SIRT6 protein level was determined by immunoblotting.

**Figure 3 ijms-25-05585-f003:**
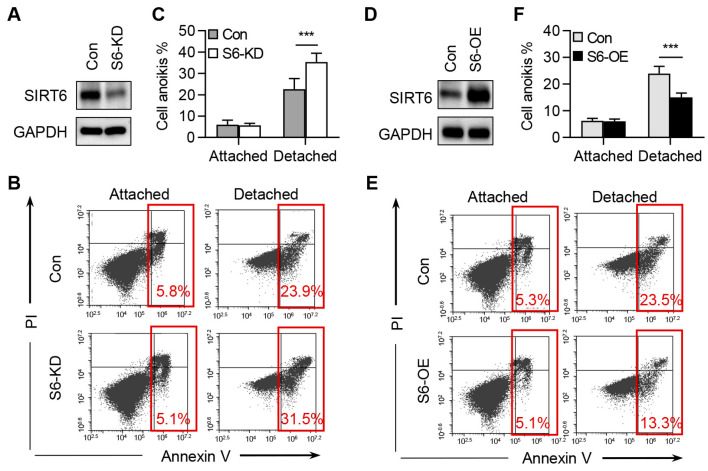
**SIRT6 inhibits the anoikis of CRC cells.** (**A**) Immunoblotting analysis of SIRT6 knockdown (S6-KD) in RKO cells. (**B**) Apoptosis levels of control and SIRT6 knockdown cells cultured in normal dishes (Attached) or ultra-low attachment dishes (Detached) for 36 h. Cell apoptosis was monitored by annexin V-FITC/PI staining and flow cytometry. (**C**) Annexin-V-positive cells in (**B**) were quantified. (**D**) Immunoblotting analysis of SIRT6 overexpression (S6-OE) in RKO cells. (**E**) Apoptosis levels of control and SIRT6 overexpression cells cultured in normal dishes (Attached) or ultra-low attachment dishes (Detached) for 36 h. Cell apoptosis was monitored by annexin V-FITC/PI staining and flow cytometry. (**F**) Annexin-V-positive cells in (**E**) were quantified. All data are presented as mean ± SD. A two-tailed *t*-test was used for statistical analysis. *** *p* < 0.001.

**Figure 4 ijms-25-05585-f004:**
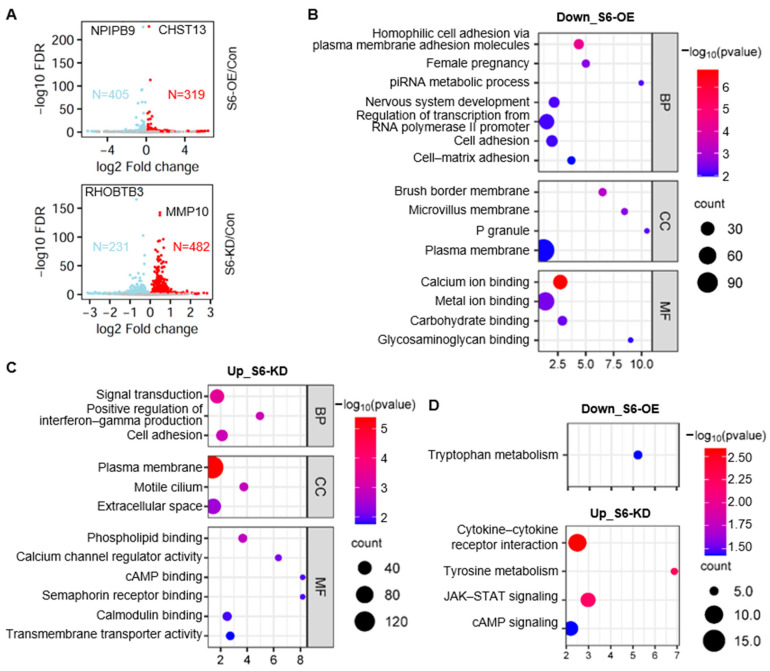
**Identification of SIRT6-regulated gene expression.** (**A**) Volcano plots of differentially expressed genes upon SIRT6 overexpression (S6-OE) or SIRT6 knockdown (S6-KD). (**B**) Gene ontology analysis of down-regulated genes upon SIRT6 overexpression. BP, biological function; CC, cellular compartment; MF, molecular function. (**C**) Gene ontology analysis of up-regulated genes upon SIRT6 knockdown. BP, biological function; cc, cellular compartment; MF, molecular function. (**D**) KEGG analysis of down-regulated genes upon SIRT6 overexpression and up-regulated genes upon SIRT6 knockdown.

**Figure 5 ijms-25-05585-f005:**
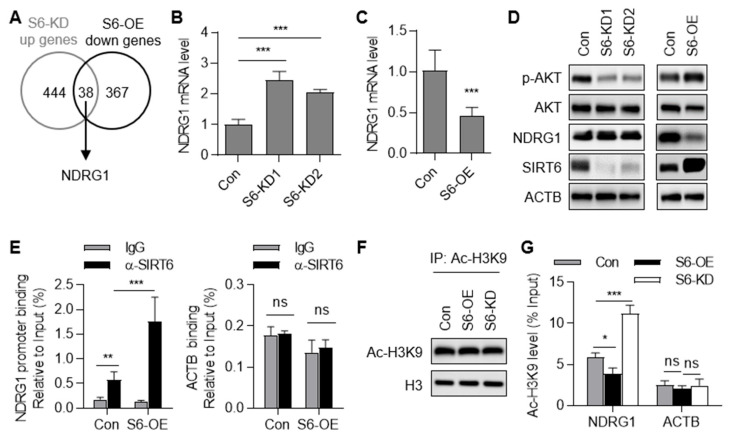
**SIRT6 represses the transcription of *NDRG1*.** (**A**) Venn diagram demonstrating the overlap of up-regulated genes in SIRT6 knockdown (S6-KD) cells and down-regulated genes in SIRT6 overexpression (S6-OE) cells. (**B**) mRNA level of *NDRG1* gene in SIRT6 knockdown cells. (**C**) mRNA level of *NDRG1* gene in SIRT6 overexpression cells. (**D**) Immunoblotting analysis of NDRG1 protein levels and AKT phosphorylation levels in SIRT6 knockdown and overexpression cells. (**E**) ChIP-qPCR analysis of SIRT6 binding to *NDRG1* gene promoter region, with *β-actin* (*ACTB*) gene serving as the control. ChIP analysis was performed with antibodies against SIRT6 or control IgG and analyzed by qPCR. Occupancies of SIRT6 in *NDRG1* gene promoter region or ACTB gene were normalized to the input DNA. (**F**) Immunoblotting analysis of Ac-H3K9 level in the immunoprecipitated products by Ac-H3K9 antibody from control (Con), SIRT6 overexpression (S6-OE), and SIRT6 knockdown (S6-KD) cells. (**G**) ChIP-qPCR analysis of acetylated histone H3K9 level (Ac-H3K9) in *NDRG1* gene promoter region. ChIP analysis was performed with antibodies against Ac-H3K9 and analyzed by qPCR. Occupancies of Ac-H3K9 in *NDRG1* gene promoter region were normalized to the input DNA. All data are presented as mean ± SD. A two-tailed *t*-test was used for statistical analysis. * *p* < 0.05, ** *p* < 0.01, *** *p* < 0.001, ns—not significant.

**Figure 6 ijms-25-05585-f006:**
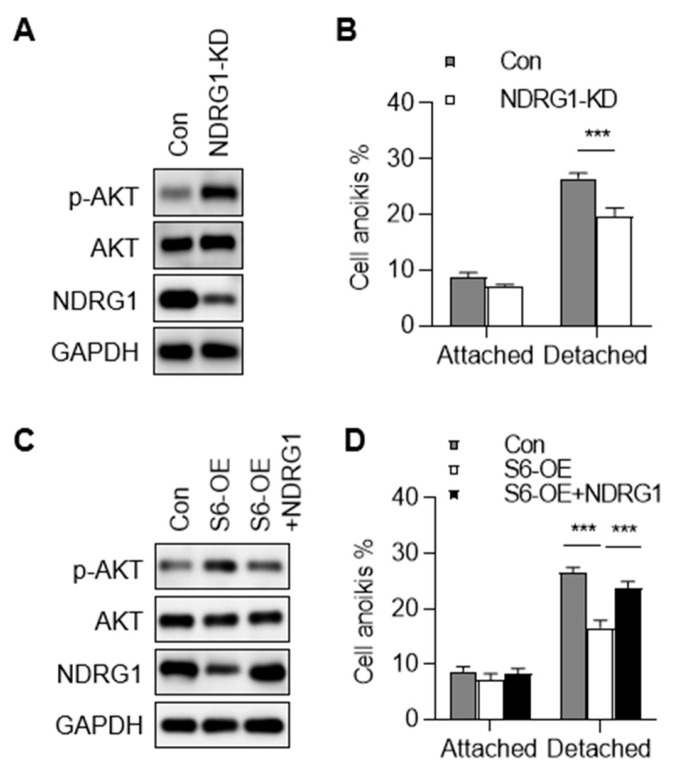
**Down-regulation of NDRG1 contributes to SIRT6-inhibited anoikis.** (**A**) Immunoblotting analysis of AKT phosphorylation level in NDRG1 knockdown (NDRG1-KD) cells. (**B**) Apoptosis levels of cells cultured in normal dishes (Attached) or ultra-low attachment plates (Detached) for 36 h. Cell apoptosis was monitored by annexin V-FITC staining and flow cytometry. Annexin-V-positive cells were quantified. (**C**) Immunoblotting analysis of AKT phosphorylation levels in control cells and SIRT6 overexpression cells with or without NDRG1 expression. (**D**) Apoptosis levels of cells cultured in normal dishes (Attached) or ultra-low attachment plates (Detached) for 36 h. Cell apoptosis was monitored by annexin V-FITC staining and flow cytometry. Annexin-V-positive cells were quantified. All data are presented as mean ± SD. A two-tailed *t*-test was used for statistical analysis. *** *p* < 0.001.

**Figure 7 ijms-25-05585-f007:**
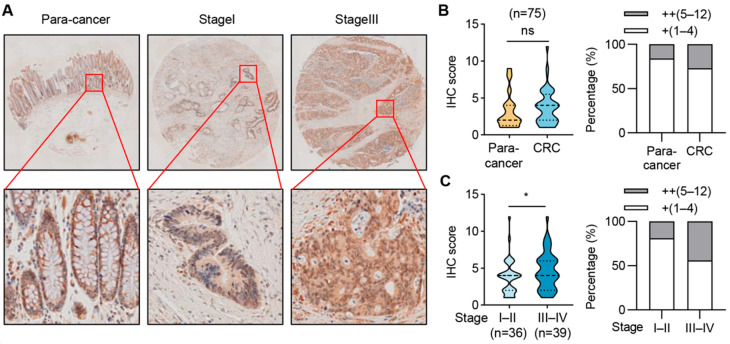
**SIRT6 expression is elevated in advanced-stage CRC samples.** (**A**) Representative immunohistochemistry (IHC) images depicting expression levels of SIRT6 protein in adjacent normal tissue, stage I CRC tissue, and stage III CRC tissue in a CRC tissue array. (**B**) IHC scoring of adjacent normal tissues and CRC tissues (left). Distribution of low (+) and high (+) levels of SIRT6 protein expression in adjacent normal tissues and CRC tissues (right). (**C**) IHC scoring of stage I–II tissues and stage III–IV tissues (left). Distribution of low (+) and high (+) levels of SIRT6 protein expression in stage I–II tissues and stage III–IV tissues (right). Statistical analysis was performed using the Chi-Square test. * *p* < 0.05, ns—not significant.

## Data Availability

Not applicable.

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
