# Peer review of "SIRT6 Inhibits Anoikis of Colorectal Cancer Cells by Down-Regulating NDRG1"

_ijms, 2024, doi:10.3390/ijms25115585_

Round 1

Reviewer 1 Report

Comments and Suggestions for Authors

Fengying Li et al. reviewed the epigenetic regulation of anoikis on cancer cell metastasis in colorectal cancer (CRC) cells. This is a good choice of topic. They found that the histone deacetylase sirtuin 6 (SIRT6) plays a pivotal role in conferring anoikis resistance to CRC cells via downregulation of N-myc downstream-regulated gene 1 (NDRG1). I believe the results are of interest. However, there are several suggestions need to be addressed before publication.

Major revisions:

1. The part of Statistical analysis in the method is lacked. In all figures, the significant difference was not analyzed.

2. Atached or detached cells were treated with cycloheximide, but the authors were not described in the method. As we know, cycloheximide can block protein synthesis, but the reason of cycloheximide administration was not introduced in this study.  

3. In Figure 1 and 3, the band of SIRT6 was different, please repeat it.  

4. Why the apoptotic rate in SW480 cells was so high, how to treat the cells.

5. In Figure 3, please provide immunofluorescence analysis of SIRT6 protein in RKO cells after SIR6 knockdown and overexpression in supplement.  

6. In Figure 5, the immunobloting of Co-IP against Ac-H3K9 should be provided.

Minor comments:

1. The abbreviations were first appeared; please provide the full name, such as KDM3A, BNIP3, and so on.

2. Why GNPDH, β-actin, and Tubulin were used.

Comments on the Quality of English Language

 Minor editing of English language required

Author Response

Responses to Reviewers

Reviewer#1:

Comments and Suggestions for Authors

Fengying Li et al. reviewed the epigenetic regulation of anoikis on cancer cell metastasis in colorectal cancer (CRC) cells. This is a good choice of topic. They found that the histone deacetylase sirtuin 6 (SIRT6) plays a pivotal role in conferring anoikis resistance to CRC cells via downregulation of N-myc downstream-regulated gene 1 (NDRG1). I believe the results are of interest. However, there are several suggestions need to be addressed before publication.

Major revisions:

  1. The part of Statistical analysis in the method is lacked. In all figures, the significant difference was not analyzed.

Response: Following the Reviewer’s suggestion, we have added a paragraph describing all the statistical analysis in the method:

Two-tailed Student’s t-test was used for the statistical analysis of qPCR data and apoptosis data. Chi-Square test was used for the statistical analysis of IHC data. P < 0.05 was regarded as statistically significant.

Moreover, we have analyzed the significant difference for all the figures.

  1. Atached or detached cells were treated with cycloheximide, but the authors were not described in the method. As we know, cycloheximide can block protein synthesis, but the reason of cycloheximide administration was not introduced in this study.

Response: To determine the degradation of SIRT6 protein, the attached or detached cells were treated with cycloheximide (CHX, 100 μg/mL) to inhibit protein synthesis for 0 hour, 4 hours, and 8 hours. The remaining SIRT6 protein level was determined by immunoblotting. We have added one paragraph in the methods part.

  1. In Figure 1 and 3, the band of SIRT6 was different, please repeat it.

Response: SIRT6 bands in Figure 1 and Figure3 are different because they use different concentration of PAGE gels. Sometimes we got smear bands for SIRT6, indicating that SIRT6 might has post-translational modifications.

    We have re-run the samples in Figure 3 using the same concentration of PAGE gel as Figure 1 and get similar pattern. We have replaced Figure 3A, 3D with the new ones.

  1. Why the apoptotic rate in SW480 cells was so high, how to treat the cells.

Response: First, SW480 cells easily undergo anoikis upon detachment. Second, we treated SW480 cells under detachment for 48 hours, while we treated RKO cells under detachment for 36 hours. We have clarified this in the manuscript.

  1. In Figure 3, please provide immunofluorescence analysis of SIRT6 protein in RKO cells after SIR6 knockdown and overexpression in supplement.

Response: Following the Reviewer’s suggestion, we have provided immunofluorescence analysis of SIRT6 protein in RKO cells after SIR6 knockdown and overexpression in supplement.

  1. In Figure 5, the immunobloting of Co-IP against Ac-H3K9 should be provided.

Response: Following the Reviewer’s suggestion, we have provided the immunoblotting of IP product against Ac-H3K9 (Figure 5F). Data revealed that the total Ac-H3K9 level did not change in SIRT6 overexpression group (S6-OE) and SIRT6 knockdown group (S6-KD) (Figure 5F).

Minor comments:

  1. The abbreviations were first appeared; please provide the full name, such as KDM3A, BNIP3, and so on.

Response: We appreciate the Reviewer’s suggestion. We have provided all the full names for the abbreviations.

  1. Why GNPDH, β-actin, and Tubulin were used.

Response: Some proteins have similar molecular weight size with GAPDH or actin. We used internal control that can be separated from the protein in the gel.

Reviewer 2 Report

Comments and Suggestions for Authors

The manuscript reads well with a good attempt made to present the role of SIRT6 in anoikis resistance to CRC cells. Below are few recommendations/feedback:

1. The introduction and the background information about the research can be explained better to avoid rephrased text.

2. It would be good to format all the gene names throughout the manuscript in italics as per the HGNC guidelines.

2. The paragraph beginning with lines 185 until 196 is italicized. Please make the formatting consistent with the other parts of manuscript unless a special reason.

3. In the methodology section - RNA sequencing (RNA-Seq analysis), please add the method of normalization, criteria for gene filtering - pval/adj value, true signal vs noise reduction etc. 

4. Similarly, for the GO analysis, please mention the filters/criteria used to select genes other than the logFC. 

5. Figure1 through 7 have the legend A in bold format, any reason why the first point is highlighted and not others?

6. Figure4 , it would be good to label the genes in the volcano plot with the maximum changes and explain the effect and participation of the these genes.

Author Response

Responses to Reviewers

Reviewer#2:

Comments and Suggestions for Authors

The manuscript reads well with a good attempt made to present the role of SIRT6 in anoikis resistance to CRC cells. Below are few recommendations/feedback:

  1. The introduction and the background information about the research can be explained better to avoid rephrased text.

Response: We have revised the introduction and the background. We think our manuscript was improved.

  1. It would be good to format all the gene names throughout the manuscript in italics as per the HGNC guidelines.

Response: We appreciate the Reviewer’s suggestion. We have formatted all the gene names (SIRT6, NDRG1, ACTB, etc.) in italics.

  1. The paragraph beginning with lines 185 until 196 is italicized. Please make the formatting consistent with the other parts of manuscript unless a special reason.

Response: We have re-formatted this part in the revised manuscript. The managing editor will finally edit all the format.

  1. In the methodology section - RNA sequencing (RNA-Seq analysis), please add the method of normalization, criteria for gene filtering - pval/adj value, true signal vs noise reduction etc.

Response: In the methodology section regarding RNA sequencing (RNA-Seq analysis), gene expression levels were normalized using the reads per kilobase million (RPKM) method to account for differences in sequencing depth and gene length. Differential genes were filtered based on statistical significance using criteria such as log2 fold change greater than 1, adjusted p-values (adj p-val) less than 0.05, and false discovery rate (FDR) less than 0.05. These thresholds ensured the selection of genes with significant differential expression while minimizing false positives. We have described these in the methods (lines126-132).

  1. Similarly, for the GO analysis, please mention the filters/criteria used to select genes other than the logFC.

Response: Differential genes for GO analysis meet the following criteria: logFC>1 or logFC<-1; adjusted p-values (adj p-val) less than 0.05, and false discovery rate (FDR) less than 0.05.

  1. Figure1 through 7 have the legend A in bold format, any reason why the first point is highlighted and not others?

Response: We appreciated the Reviewer’s suggestion. Actually, the manuscript was formatted by the managing editor. Instead of legend A, we have formatted the figure legend time in bold in the revised manuscript.

  1. Figure4 , it would be good to label the genes in the volcano plot with the maximum changes and explain the effect and participation of the these genes.

Response: Following the Reviewer’s suggestion, we have labeled the genes in the volcano plot with maximum changes in Figure 4A. We have also described these genes in the revised manuscript (lines205-212).

Round 2

Reviewer 1 Report

Comments and Suggestions for Authors

no